# Electrosprayed Nanoparticles Containing Mangiferin-Rich Extract from Mango Leaves for Cosmeceutical Application

**DOI:** 10.3390/nano13222931

**Published:** 2023-11-11

**Authors:** Vissuta Sirirungsee, Pawitrabhorn Samutrtai, Padchanee Sangthong, Phakorn Papan, Pimporn Leelapornpisid, Chalermpong Saenjum, Busaban Sirithunyalug

**Affiliations:** 1Master’s Degree Program in Cosmetic Science, Faculty of Pharmacy, Chiang Mai University, Chiang Mai 50200, Thailand; vissuta_s@cmu.ac.th; 2Department of Pharmaceutical Sciences, Faculty of Pharmacy, Chiang Mai University, Chiang Mai 50200, Thailand; pawitrabhorn.s@cmu.ac.th (P.S.); pim_leela@hotmail.com (P.L.); 3Department of Chemistry, Faculty of Science, Chiang Mai University, Chiang Mai 50200, Thailand; padchanee.sangthong@cmu.ac.th (P.S.); phakorn_papan@cmu.ac.th (P.P.); 4Innovation Center for Holistic Health, Nutraceuticals, and Cosmeceuticals, Faculty of Pharmacy, Chiang Mai University, Chiang Mai 50200, Thailand; 5Research Center for Innovation in Analytical Science and Technology for Biodiversity-Based Economic and Society, Chiang Mai University, Chiang Mai 50200, Thailand

**Keywords:** mangiferin, aging, mango leaves, nanoparticles, electrospraying, cosmeceuticals

## Abstract

Mango (*Mangifera indica* L.) is one of the most economically important fruits in Thailand. Mango has been used as a traditional medicine because it possesses many biological activities, such as antioxidant properties, anti-inflammatory properties, microorganism-growth inhibition, etc. Among its natural pharmacologically active compounds, mangiferin is the main active component found in mango leaves. Mangiferin has the potential to treat a variety of diseases due to its multifunctional activities. This study aims to prepare a mangiferin-rich extract (MRE) from mango leaves and develop nanoparticles containing the MRE using an electrospraying technique to apply it in a cosmeceutical formulation. The potential cosmeceutical mechanisms of the MRE were investigated using proteomic analysis. The MRE is involved in actin-filament organization, the positive regulation of cytoskeleton organization, etc. Moreover, the related mechanism to its cosmeceutical activity is metalloenzyme-activity regulation. Nanoparticles were prepared from 0.8% *w*/*v* MRE and 2% *w*/*v* Eudragit^®^ L100 solution using an electrospraying process. The mean size of the MRE-loaded nanoparticles (MNPs) received was 247.8 nm, with a PDI 0.271. The MRE entrapment by the process was quantified as 84.9%, indicating a high encapsulation efficiency. For the skin-retention study, the mangiferin content in the MNP-containing emulsion-gel membranes was examined and found to be greater than in the membranes of the MRE solution, illustrating that the MNPs produced by the electrospraying technique help transdermal delivery for cosmetic applications.

## 1. Introduction

Mango (*Mangifera indica* L.) is one of the most economically important fruits in Thailand. According to the information from the Trade Policy and Strategy Office of Thailand, Thailand exported around 113,806 tons of mangos, valued at THB 2.9 billion, in 2021. The expanding rate jumped to 50.25% higher than the previous month [1]. Mango is a stone fruit found in tropical and subtropical regions. It constitutes one of the most commercialized fruits in the world, belonging to the Anacardiaceae family [2]. It has long been used in Ayurvedic medicine, and different parts of the mango are used as an antiseptic, dentifrice, and diuretic, and to treat diarrhea, dysentery, asthma, hypertension, and toothache [3]. Mango possesses many biological activities, such as antioxidant, anti-inflammatory, antiobesity, anticancer activity, microbial-growth inhibition, etc. [4]. One related study reported that mango is composed of carotenoids, anthocyanins, dietary fiber, and phenolic compounds [5]. Phenolic compounds in mango include mangiferin, gallic acid, caffeic acid, and tannin [6].

One of the most bioactive compositions found in the leaves, bark, root, peel, and flesh of mango is mangiferin. Mangiferin, generally called C-glycosyl xanthone, is a xanthone connected to a C-glycoside (structure shown in Figure 1), displaying numerous biological actions contributing to therapeutic, pharmaceutical, and cosmetic applications [7]. Its biological activities include antioxidant effects, which are thought to have a greater potential than vitamins C and E [8], antiviral, anticancer, antidiabetic, antiaging, immunomodulatory, hepatoprotective, and analgesic effects [9]. Regarding its antioxidant activity, methanolic extract, which consists of mangiferin, exhibited a potent antioxidant activity (EC_50_ = 5.8 ± 0.96 µg/mL) that was comparable to rutin (EC_50_ = 5.56 ± 0.33 µg/mL) [10]. Mangiferin could reduce hydrogen peroxide-induced lipid peroxidation in human peripheral blood lymphocytes in a concentration-dependent manner, and inhibited the induction of OH^•^, O_2_^−^, DPPH, and ABTS radicals in a dose-dependent manner in cell-free studies. It demonstrated that mangiferin has radioprotective properties by suppressing the effects of free radicals [11]. The antioxidant activity of mangiferin prevented UVB-induced wrinkle formation by reducing the UVB-induced matrix metalloproteinase (MMP-9) protein expression and enzyme activity, and attenuating the UVB-induced phosphorylation of mitogen-activated protein kinase 1 (MEK) and extracellular-signal-regulated kinase (ERK). In one in vivo study, mangiferin inhibited UVB-induced skin wrinkles, epidermal thickening, and damage to collagen, suggesting mangiferin exerts antiphotoaging [12]. Basically, the skin dry weight is mainly composed of 70–80% collagen, which provides strength to the skin. Elastin is involved in skin elasticity, accounting for 2–4% of the extracellular matrix (ECM), while glycosaminoglycans (GAGs), such as hyaluronic acid, are responsible for the hydration of the skin. Collagen, elastin, and GAGs are proteins produced by fibroblasts, which can be affected by photoaging, leading to wrinkles, hyperpigmentation, and skin thickness.

Electrospraying is a technique producing nano/microparticles by applying an electric field that has been widely used in food as well as the nutraceutical industry, etc. The electrospraying setup consists of a high-voltage supplier, a syringe-loading polymer solution, a needle equipped with a syringe, a syringe pump, and a collector [13]. When applying an electric field, the high voltage causes the polymer solution to become charged, and the charged solution is ejected from the needle. The density of charged droplets increases, so that repulsive force of the charge surpasses the surface tension, resulting in a “Taylor cone”. After evaporating the solvent, the generated nanoparticles are sprayed onto a collector. The electrospraying technique is a simple and inexpensive technique to set up. It provides a high encapsulation efficiency (EE) and can be used with high-molecular-weight polymers [14]. According to its simplicity and versatility, the electrospraying technique is an effective option for drug delivery.

Proteomics is the study of the interactions, functions, composition of the organism, and structures of proteins and their cellular activities. It provides a better understanding of the structure and function of the organism than genomics. The study of proteomics has many applications, such as medicine, oncology, food microbiology, and agriculture. Proteomics is a fast, sensitive technology providing high proteome coverage [15]. It helps discover new biomarkers in different diseases, the mechanisms of drugs, as well as the cosmeceutical mechanisms of the MRE.

After harvesting, mango leaves are discarded as agricultural waste. Thus, this study aims to prepare the mangiferin-rich extract (MRE) from mango “Talapnak”-variety leaves and investigate its antioxidant, antiaging, and antityrosinase properties. The electrospraying technique was used to produce nanoparticles. Nanoparticles containing the MRE were then formulated into a cosmetic product. Additionally, proteomics will provide more information of a possible mechanism between the MRE and proteins associated to antiaging activity.

## 2. Material and Methods

### 2.1. Materials

Mango “Talapnak”-variety leaves were sourced from Banpan District, Lamphun Province, Thailand. DPPH (1,1-diphenyl-2-picryhydrazyl), mangiferin, Trolox, linoleic acid, N-succinyl-AlaAlaAla-p-nitroanilide (AAAPVN), L-tyrosine, collagenase from *Histidium histolyticum*, elastase from porcine pancreas, hyaluronidase from bovine, tyrosinase from mushroom, EGCG (epigallocatechin gallate), tannic acid, hyaluronic acid sodium salt, activated charcoal, and collagen were obtained from Sigma-Aldrich, St. Louis, MO, USA. Ascorbic acid and N,N-dimethylacetamide were purchased from Lobachemie Pvt. Ltd., Mumbai, India. Eudragit^®^ L100 was obtained from Evonik, Darmstadt, Germany. Isopropanol, ethanol, H_2_SO_4_, and n-hexane were bought from RCI Labscan, Bangkok, Thailand. All solvents were of analytical grade.

### 2.2. Mango Leaf Extraction

Mango “Talapnak”-variety leaves were cleaned and dried in a hot-air oven at 50 °C for 24 h. The dried mango leaves were pulverized into powder. Mango leaf powder was then extracted according to the method of Vo et al. (2017) with modifications [16]. In all, 200 g of the mango leaf powder was extracted using 1600 mL of 50%, 60%, or 70% isopropanol (IPA) and using reflux extraction for 1 h. Then, the solution was cooled and filtered. After evaporating to 400 mL, the solution was partitioned with 400 mL of n-hexane three times, adjusted to a pH of 2 with 1 M H_2_SO_4_, and boiled in a reflux condenser for 1 h. The solution was evaporated to 80 mL. Exactly 1 g of charcoal and 80 mL of dioxane were added to the solution. The solution was then filtered before storing in a refrigerator for 24 h; the precipitate was filtered and washed with 60 mL of cold ethanol three times and dried in a vacuum oven at 70 °C for 4 h. The obtained precipitate is called MRE. The MRE extraction was performed ten times to obtain an adequate amount of the MRE. To purify the mangiferin, the MRE was dissolved in 50% IPA, then boiled while stirring for 5 min. The solution was then filtered and stored in a refrigerator for 24 h. The precipitate was filtered and washed with cold ethanol and dried in a vacuum oven, thus obtaining purified MRE (pMRE).

### 2.3. Characterization of the MRE and pMRE

#### 2.3.1. UV–Vis Spectrophotometer

The MRE and pMRE were characterized for mangiferin using a UV–Vis spectrophotometer (Shimadzu UV2600i, Tokyo, Japan) by preparing 0.001% of the extract in methanol. The scanning wavelength was from 200 to 800 nm.

#### 2.3.2. Fourier Transform Infrared Spectrometer (FT-IR)

An amount of 1 mg of the pMRE was measured using the potassium–bromide (KBr) pellet method in an FI-IR spectrometer. The resolution was 4.0 cm^−1^. The spectral region was between 4000 and 400 cm^−1^, and the result was compared with the reference standard mangiferin.

#### 2.3.3. Differential Scanning Calorimetry (DSC)

The MRE was placed in a standard aluminum pan and covered. Measurement was performed from 25 to 300 °C at a constant heating rate of 10 °C/min in a dynamic nitrogen atmosphere (flow rate = 30 mL/min).

### 2.4. Standardization of the MRE

The mangiferin amount was determined using high-performance liquid chromatography (HPLC). Briefly, the HPLC system (Agilent, Santa Clara, CA, USA) with a C18 analytical column was used with a detection wavelength of 258 nm. The mobile phase was composed of 2% acetic acid (A) and 0.5% acetic acid: acetonitrile (1:1 *v*/*v*) (B). The gradient elution was performed as follows: 0 to 2 min, 5% of B; 2 to 10 min, 5 to 25% of B; 10 to 40 min, 25 to 55% of B; 40 to 45 min, 55 to 90% of B; 45 to 50 min, 90 to 55% of B, 50 to 55 min, 55 to 5% of B; 55 to 60 min, 5% of B. The injection volume, the flow rate, and the column temperature were 20 µL, 1.0 mL/min, and 25 °C, respectively. The mangiferin amount was quantified by comparing the peak area of the sample to the standard curve. The mangiferin stock solution of 500 ppm was diluted with 50% IPA to concentrations of 25, 50, 75, 100, and 125 ppm.

### 2.5. Bioactivities Assessment of the MRE

#### 2.5.1. Antioxidant Activity

##### DPPH Radical Scavenging Assay

Antioxidant activity was determined by preparing various concentrations of the MRE. In a 96-well plate, 20 µL of the samples were mixed with 180 µL of the methanolic DPPH solution. The mixture was incubated for 30 min at RT in the dark. The absorbance was read at 520 nm using a microplate reader (BMG Labtech, Ortenberg, Germany), and ascorbic acid was used as a positive control. The % inhibition was calculated using the equation below:$$\%\;\mathrm{Inhibition}=\frac{A_{c}-A_{s}}{A_{c}}\times 100$$
where A_c_ and A_s_ were the absorbances of the control and sample, respectively. The IC_50_ value was calculated from the polynomial equation of the plot between the % inhibition and concentration. The samples were performed in triplicate, and the average IC_50_ value was reported as mean ± SD.

##### Lipid Peroxidation Inhibition Assay

The linoleic acid peroxidation method was used to determine the lipid peroxidation inhibition. Various concentrations of the MRE were prepared. In Eppendorf tubes, 75 µL of the samples/solvent, 350 µL of 1.3% *w*/*v* of linoleic acid in methanol, 175 µL of DI water, 350 µL of 20 mM phosphate buffer with a pH of 7.0, and 50 µL of 46.35 mM 2,2′-Azobis(2-aminopropane) dihydrochloride (AAPH) were mixed. The mixtures were incubated at 50 °C for 4 h. Then, 5 µL of the mixture was added to a 96-well plate. After adding 5 µL of 10% NH4SCN, 5 µL of 20 mM FeCl2, and 185 µL of 75% methanol, the 96-well plate was incubated at RT for 5 min; then, the absorbance was measured at 500 nm using a microplate reader (BMG Labtech, Ortenberg, Germany). Trolox was used as a positive control.

#### 2.5.2. Antiaging Activity

##### Anticollagenase Assay

Anticollagenase activity was investigated by mixing the samples with collagenase enzyme, 5 mM CaCl_2_ in borate buffer with a pH of 7.5, and DI water. After incubating at 37 °C for 10 min, 1% collagen solution was added. The mixture was incubated at 37 °C for 1 h. Then, 0.75 mM of 3,4-DHPAA, 125 mM Borate buffer with a pH of 7.5, and 1.25 mM sodium periodate were added and incubated at 37 °C for 10 min. The mixture was maintained in an ice bath. The reaction was read using a microplate reader with the fluorescence mode at 375 and 465 nm. The positive control was ascorbic acid.

##### Antihyaluronidase Assay

The antihyaluronidase assay was performed according to Chaiyana et al. (2020) with some modifications [17] in Eppendorf tubes. The samples were mixed with 2 mg/mL of hyaluronidase enzyme and incubated for 10 min at 37 °C. Then, 0.03% hyaluronic acid was added before incubating at 37 °C for 45 min. After that, an acidic albumin solution with a pH of 3.4 was mixed and incubated at RT for 10 min. Then, 200 µL of the mixtures were pipetted in a 96-well plate, and the absorbance was read at 600 nm. Tannic acid was used as a positive control.

##### Antielastase Assay

According to the method of Theansungnoen et al. with modifications [18], the assay was performed in 0.2 M Tris–HCl buffer (pH of 8.0). Then, 0.04 U/mL of porcine pancreatic elastase was prepared in sterile water. After that, 1 mg/mL of N-Succinyl-Ala-Ala-Ala-*p*-nitroanilide (SANA) was used as a substrate. The sample solutions were mixed with elastase enzyme and Tris–HCl buffer before measuring the reaction spectrophotometrically at 410 nm using a microplate reader (BMGlabtech, Ortenberg, Germany) with the kinetic mode. EGCG represents a positive control.

#### 2.5.3. Antityrosinase Assay

The inhibition of the tyrosinase activity was measured using L-tyrosine as a substrate, and the positive control was ascorbic acid. Tyrosinase enzyme, mixing with the samples, and 20 mM phosphate buffer with a pH of 6.8 were incubated at RT for 10 min. Then, the substrate was added and incubated for another 20 min at RT. The absorbance was read at 492 nm using a microplate reader. The % tyrosinase inhibition was calculated using the equation as mentioned above.

### 2.6. Proteomics Analysis of the MRE

#### 2.6.1. Cell Preparation and Protein Extraction

To investigate protein expression of the MRE, 1 × 10^5^ cells/mL of mouse fibroblast cells (L929) was seeded in a T-25 flask with RPMI 1640 medium containing 10% FBS and 1% penicillin/streptomycin. The cells were treated with the MRE at the concentration of 31.25 µg/mL for 24 h. After harvesting by directly scraping in sterile cold PBS, the cells were collected and centrifuged at 14,000 rpm for 20 min before washing with sterile cold PBS three times. The pellet was snap-frozen rapidly in liquid nitrogen and stored at −80 °C until the proteomic analysis.

#### 2.6.2. LC-MS/MS Analysis

The harvested cells were lysed using lysis buffer (0.5% SDS, 5 mM DTT in 0.1× PBS, pH of 7.4) with 1 mm PMSF for the LC-MS/MS before being subjected to tandem–mass spectroscopy using a nano-LC system coupled with a high-resolution 6600 TripleTOF^TM^ (AB-Sciex, Concord, ON, Canada). The conditions were as follows: mobile phase A consisted of 0.1% formic acid in water and mobile phase B comprised 95% acetonitrile with 0.1% formic acid. The LC parameters were composed of a 135 min long process for a single injection, and the analytical temperature was set at 55 °C. The MS scanned a mass ranging from 400 to 1600 m/z using the data-dependent acquisition mode. The top 20 most abundant peptide ions with charge states ranging from 2 to 5 (positive mode) for fragmentation were selected, and the dynamic-exclusion duration was set at 15 s.

#### 2.6.3. Protein Identification

The raw data of the MS-spectra file were extracted, and protein sequences were annotated using the Paragon^TM^ Algorithm by ProteinPilot^TM^ Software. RStudio ver. 2023.03.0 Software was applied to analyze the MS data to identify proteins. Those proteins with a log_10_ (*p* value) greater than 1.3 were considered significant. The proteins with log_2_ fold changes less than −0.5 and higher than 0.5 were classified as downregulated and upregulated, respectively. Downregulation refers to a decrease in the production of a protein, while upregulation describes an increase in the production of a protein. In proteomics, downregulation and upregulation can be used to study the effects of stimuli on protein expression. STRING analysis was used to further investigate the interaction of these identified upregulated proteins with a high interaction value (0.7); the false discovery rate was set at less than 0.05.

### 2.7. Preparation of MRE-Loaded Nanoparticles (MNPs) by the Electrospraying Technique

Spraying solution was prepared by dissolving 2% *w*/*v* of Eudragit^®^ L100 in ethanol and 0.8% *w*/*v* of the MRE in dimethylacetamide at ratio of 9:1. The polymer was stirred using a magnetic stirrer at RT until clear. To obtain an appropriate condition, the parameters, including the applying voltage (kV), flow rate (mL/min), and the distance between the tip and collector (cm), were varied. A foil-covered aluminum plate was used as a collector. The electrospraying process was performed by connecting the positive electrode of the voltage power supply to a metal needle tip, while the ground electrode was connected to the collector plate. The polymer solution was pumped through a 10 mL syringe with a 1.20 mm diameter 18G needle. The electrospraying was carried out under 30 to 40% RH. The particle size, polydispersity index (PDI), and zeta potential were studied in deionized water using a Malvern zetasizer (Malvern Instrument Ltd., Malvern, UK). Particle morphology was investigated using a JSM-IT300 scanning electron microscope (JEOL Ltd., Boston, MA, USA). The %EE was adapted with modifications [19] by dissolving the MNPs and MRE in a suitable solvent. The suspensions were centrifuged at 10,000 rpm for 30 min. Supernatants were filtered through a 0.22 micron filter, and the %EE was evaluated by HPLC following the formula below:$$\%\;\mathrm{Entrapment}\;\mathrm{efficiency}\;(\mathrm{EE}\%)=\frac{Amount\;of\;drug\;in\;nanocarriers}{Initial\;amount\;of\;drug}\times 100$$

### 2.8. Irritation Study

The irritation study of the MRE and MNPs were conducted using the hen’s egg chorioallantoic membrane (HET-CAM) test according to Yeerong et al. (2021) [20]. The HET-CAM was prepared from 7 to 9 days using fertilized eggs. The air sac was cut with a rotating dentist saw blade; then, the eggshell was slowly peeled off. After adding normal saline solution, the eggs were incubated for 15 min. The inner-eggshell membrane was then removed with forceps, and 30 µL of the tested solution was applied. The reaction was observed for 5 min. Eggs treated with 1% *w*/*v* sodium lauryl sulfate (SLS) were set up as the positive control, and eggs treated with the 0.9% NaCl solution served as the negative control. The time to the onset of vascular hemorrhage, vascular lysis, and vascular coagulation, as well as the irritation score (IS), were observed using the formula below:

$$\mathrm{IS}=\frac{\left(301-t\left(h\right)\right)}{300}\;\times \;5+\frac{\left(301-t\left(l\right)\right)}{300}\;\times \;7+\frac{\left(301-t\left(c\right)\right)}{300}\times \;9$$
where t (h) = the time to the onset of vascular hemorrhage within 300 s after applying the test substance (s); t (l) = the time to the onset of vascular lysis within 300 s after applying the test substance (s); t (c) = the time to the onset of vascular coagulation within 300 s after applying the test substance (s). The irritation score (IS) was calculated and expressed as mean ± SD, and the test was performed in replicate. A score from 0.0 to 0.9 indicated no irritation, 1.0 to 4.9 indicated weak irritation, 5.0 to 8.9 indicated moderate irritation, and 9.0 to 21.0 indicated severe irritation.

### 2.9. Preparation of the Formulation Containing MNPs

Table 1 demonstrates the ingredients of the emulsion gel. Carbopol 940 was dispersed in DI water and stirred to form a gel. Then, glycerin was added to the gel base, and light-cream maker and jojoba oil were mixed. Cyclopentasiloxane and dimethicone cross-polymer (DC9045) was dissolved in cyclopentasiloxane and dimethicone, and then introduced in the gel base. Disodium EDTA dissolving in DI water and DMDM hydantoin were added, and the emulsion gel was adjusted to a pH from 5.5 to 6 with triethanolamine.

The stability test of the formulations was evaluated by heating–cooling cycling. The emulsion gel base and emulsion gel containing MNPs were stored at 45 °C for 24 h and 4 °C for 24 h. Heating–cooling cycles were performed for six cycles.

### 2.10. Skin-Retention Study

Franz cell diffusion was performed to determine the permeability of the MRE solution and the emulsion gel containing the MNPs. The MNP-containing emulsion gel and the MRE solution were applied to the Strat-M^®^ membrane (diameter 25 mm, thickness 300 µm). The membranes were mounted on Franz cells (Logan Instruments Corp, Somerset, NJ, USA), and the diffusion area was 1.76 cm^2^. The receiver compartment was filled with PBS with a pH of 7.4 with a constant temperature of 32.0 °C. Then, 1 g of the MNP-loaded emulsion gel was placed in the donor compartment. Samples were taken from the receiver compartment at intervals of 30 min, and 1, 3, 6, 9, and 12 h, to analyze the mangiferin amount. After each interval, the receptor part was replaced with the same amount of fresh PBS. The membranes were cut into pieces and filled with solvent, then sonicated for 15 min. The samples were filtered through a 0.22 µm membrane filter for the HPLC analysis of the quantity of mangiferin.

### 2.11. Statistical Analysis

The results have been described as mean ± SD of three replicates. All statistical analyses were calculated using SPSS Software, Version 17.0, using a one-way ANOVA or the t-test at the 5% level (*p* < 0.05).

## 3. Results and Discussion

### 3.1. MRE Yield and Characterization

The result of the MRE yield depends on the ratio of solvents used. Extraction of mango “Talapnak” leaf powder with 50, 60, and 70% isopropanol by reflux extraction yielded 0.618 ± 0.054%, 0.881 ± 0.593%, and 1.073 ± 0.093%, respectively. The appearance of the extract was a fine pale-yellow powder.

The use of 70% isopropanol provided the greatest yield compared with the other ratios. Therefore, the extract from 70% isopropanol was chosen for further investigation. The purification process was performed to confirm the presence of mangiferin in the MRE. The pMRE yielded 0.042% with a 77.4% purity. The result of the characterization of the MRE and pMRE using a UV–Vis spectrophotometer showed absorption peaks at 241, 257.5, 315, and 365 nm, as shown in Figure 2a,b, which was related to absorption peaks presented in Vietnamese pharmacopeia in Figure 2c [21].

FT-IR was used to identify the chemical structure of the pMRE. Figure 3 demonstrates functional groups of mangiferin at the peaks 3370.08 cm^–1^ (O–H str), 2938.97 cm^–1^ (C–H str), 1649.67 cm^–1^ (C=O str), 1622.44 cm^–1^ (C=C str), 1494.06 cm^–1^ (CH–CH str), 1254.49 cm^–1^ (C–O str), and 1095.56 cm^−1^ (O–H deformation), which was consistent to relevant studies [16,22]. This result confirmed the presence of the mangiferin molecule in the MRE.

The DSC was investigated for the thermal transition of a material. Figure 4 depicts the DSC thermogram, revealing an endothermic peak of the MRE at 259.8 °C compared with the reference standard mangiferin at 272 °C. It could have been caused by the MRE containing other compounds, which lower those of the endothermic peak of the reference mangiferin.

Basically, pure substance melts at a sharp and highly defined temperature. On the other hand, contaminated substances typically display a large melting point range, and lower than that of a pure substance, due to melting-point depression. Because the impurities disrupt the crystal structure of the pure substance, resulting in the weaker of intermolecular forces, the melting point of the MRE displayed as lower than the reference mangiferin.

### 3.2. Standardization of the MRE

The mangiferin content in the samples was standardized using HPLC, as shown in Figure 5. Three of the ten MRE samples extracted with 70% IPA were chosen randomly to investigate the mangiferin content. The result demonstrated no significant difference among the samples (*p* > 0.05), as shown in Table 2. Therefore, each sample contained mangiferin over 50%, which was largely present in the extracts.

### 3.3. Bioactivities Assessment of the MRE

#### 3.3.1. Antioxidant Activity

The DPPH radical scavenging assay and lipid peroxidation inhibition assay were conducted to measure the antioxidant activity of the MRE. The IC_50_ on the DPPH radical scavenging activity of the MRE was 4.967 ± 0.588 µg/mL, which differed insignificantly from the standard mangiferin (4.780 ± 1.147 µg/mL) and ascorbic acid (5.676 ± 0.398 µg/mL). According to Masibo and He (2008), mangiferin was more effective than ascorbic acid, which is related to the result [8]. On the other hand, the MRE at a concentration of 4.5 mg/mL exhibited a 45.673 ± 0.944% inhibition of lipid peroxidation. Nevertheless, Trolox and ascorbic acid at a concentration of 4.5 mg/mL could inhibit 98.005 ± 0.001% and 99.282 ± 0.406%, respectively. As a result, the MRE scavenged the DPPH radical better than it inhibited lipid peroxidation according to the four hydroxyl groups of mangiferin, which can donate hydrogen atoms to free radicals [23,24]. Furthermore, mangiferin is a phenolic compound that comprises a hydrophilic antioxidant. For this reason, the DPPH assay is a better choice for testing the antioxidant activity of mangiferin than the lipid peroxidation assay.

#### 3.3.2. Anticollagenase Activity

Collagen is a structural protein containing amino acids in the dermis, accounting for 70 to 80% of the dry weight of the skin, and helping to strengthen, hydrate, and provide elasticity to the skin. However, as we age, the imbalance between collagen production and collagen breakdown leads to a loss of collagen. The IC_50s_ of the MRE, standard mangiferin, and ascorbic acid on collagenase inhibition were 53.63 ± 0.002 µg/mL, 30.06 ± 0.001 µg/mL, and 107.84 ± 0.001 µg/mL, respectively, as depicted in Figure 6a. The IC_50_ of the MRE significantly differed from that of the standard mangiferin and ascorbic acid (*p* < 0.05). Additionally, the MRE exhibited excellent collagenase inhibition activity. The result from the study of Ochocka et al. (2017) revealed that standard mangiferin could inhibit collagenase in a dose-dependent manner [25], which was comparable and consistent to our study.

#### 3.3.3. Antihyaluronidase Activity

Hyaluronic acid is a glycosaminoglycan found in the skin that retains moisture content. It helps to hydrate the skin. Hyaluronic acid can be hydrolyzed by hyaluronidase. The MRE at a concentration of 40.0 µg/mL inhibited hyaluronidase by 2.25 ± 1.84%, while the standard mangiferin at 40.0 µg/mL could inhibit it by 1.72 ± 1.13%. Tannic acid, a positive control, exhibited 90.61 ± 1.04% of antihyaluronidase activity, as represented in Figure 6b. The % inhibition of the MRE and standard mangiferin on hyaluronidase inhibition significantly differed from tannic acid (*p* < 0.05). As a result, the MRE showed a very low inhibitory activity in the antihyaluronidase assay. Surprisingly, the previous study indicated a strong inhibition on hyaluronidase of the mangiferin extracts from honeybush [26]. This may be caused by using different plant sources and solvents in the isolation process, which led to the presence of substances antagonizing the antihyaluronidase effect of mangiferin.

#### 3.3.4. Antielastase Activity

Elastase is an enzyme that degrades elastin, an extracellular matrix that provides elasticity to tissue. Figure 6c illustrates that EGCG, a positive control, showed excellent activity at 0.7 mg/mL, with 74.97 ± 0.82% elastase inhibition. Regarding the antielastase of the MRE and standard mangiferin, the percentage of elastase inhibition was 47.93 ± 1.34% and 43.57 ± 0.61%, respectively. The EGCG showed a significant difference compared to the MRE and standard mangiferin (*p* < 0.05). Our result was comparable to the prior study [26].

#### 3.3.5. Antityrosinase Activity

The tyrosinase enzyme is involved in the melanogenesis pathway. This enzyme catalyzes tyrosine to melanin and other pigments. The IC_50_ of the MRE with the L-tyrosine substrate was 877.49 ± 0.063 µg/mL. Ascorbic acid showed an IC_50_ of 69.8 ± 0.018 µg/mL (Figure 6d). A significant difference was found between the IC_50_ of the MRE and that of ascorbic acid (*p* < 0.05). The result demonstrated that the MRE inhibited the tyrosinase enzyme, but not as effectively as ascorbic acid. The previous study demonstrated that the standard mangiferin had a stronger tyrosinase inhibition activity than the extract (IC_50_ = 195.50 ± 1.40 µg/mL), but not as much as kojic acid (IC_50_ = 24.65 ± 0.61 µg/mL). The inhibitory action of mangiferin may involve the carbonyl and phenolic groups in the structure, as it is a copper chelator. Because tyrosinase is a metalloenzyme, the copper chelator can inhibit tyrosinase by mimicking the substrate at the active sites [27].

### 3.4. Proteomics Analysis of the MRE

The proteomics of the MRE were analyzed using R-studio. Altogether, 779 proteins were found. Mostly, protein expression changes were downregulated (91.2%; 711 of 779), while 54 proteins were not significant (6.9%). The cutoff on the axis was chosen arbitrarily to identify the upregulated proteins. Among the 779 proteins, almost 14 proteins were observed to be upregulated (0.8%; 14 of 779), as shown in Figure 7. The description and function of these proteins are demonstrated in Table 3.

These proteins are involved in actin-filament organization, cytoskeleton organization, Rho protein signal transduction, the regulation of actin-filament polymerization, cytoplasmic translation, muscle contraction, the positive regulation of supramolecule fiber organization, the translation and regulation of actin-cytoskeleton organization, the regulation of cytoskeleton organization, and the regulation of catalytic activity. It has been suggested that the actin cytoskeleton is essential for an arrangement of the collagenous extracellular matrix in embryonic tendons [28]. Actin disassembly reduced the TGF-β pathway, regulating collagen production in dermal fibroblasts [29]. Excess collagen production results in fibrosis, where mangiferin that has been shown to reduce BLM-induced pulmonary fibrosis in mice by blocking the TLR4/p65 and TGF-β1/Smad2/3 pathways, perhaps preventing pulmonary fibrosis in mice [30,31]. Another finding demonstrated that mangiferin decreased RAC1/WAVE2, belonging to the Rho proteins, assuming an important role in cytoskeletal rearrangement and promoting malignant cell migration and invasion, resulting in the inhibition of breast tumor progression [32]. Regulating the actin-filament polymerization in the nucleus of mangiferin causes the regulation of gene transcription and repairs damaged DNA. According to prior research, mangiferin reduced DNA damage and improved DNA repair in human peripheral blood lymphocytes exposed to 1–4 Gy γ-radiation [33]. The result of the MRE on collagenase and elastase activity inhibition is associated with regulating catalytic or metalloenzyme activity. One study examined that mangiferin reversibly inhibited elastase and collagenase in a noncompetitive manner. Mangiferin interacts with the free form of enzymes at a location other than the active site, and the enzyme–substrate complex [25]. The catalytic-activity regulation of the MRE is related to the antiaging activity. In addition, mangiferin is a potent antioxidant and probably reduces the activity of the enzyme by controlling free radicals induced by UV radiation.

### 3.5. Evaluation of MNPs

The 2% Eudragit^®^ L100 was dissolved in ethanol. In all, 0.8% of the MRE in dimethylacetamide was added to the polymer solution. The appropriate condition involved applying voltage of 12 kV with 30 cm of distance between the needle tip and the collector at flow rate of 0.7 mL/min. The mixture was electrosprayed to create nanoparticles with a mean size of 247.8 nanometers and a polydispersity index (PDI) of 0.271. This indicates that the nanoparticles were homogenous in size. The zeta potential of the particles was −31.7 mV. Then, the % entrapment efficiency of the MNPs was 84.9%, and the morphology of nanoparticles was evaluated using SEM images, as shown in Figure 8.

### 3.6. Irritation Study

To evaluate the irritation potential of the MRE and MNPs, the hen’s egg chorioallantoic membrane (HET-CAM) test was used. The result showed that the 1% *w*/*v* SLS, a positive control, caused severe irritation, as displayed in Table 4. Vascular hemorrhage, vascular lysis, and vascular coagulation were onset within 5 min after application. The irritation score of 1% *w*/*v* SLS was 14.8 ± 2.2, describing severe irritation. Moreover, the 0.9% *w*/*v* NaCl, 0.5 mg/mL of MNP, and 0.5 mg/mL of the MRE were observed to be nontoxic, with an IS 0.0 ± 0, indicating no irritation. Overall, the MRE and MNPs are safe to use as an active ingredient in cosmetic formulations.

### 3.7. Preparation and Evaluation of the Emulsion Gel Containing MRE-Loaded Nanoparticles

The physical appearance of the emulsion gel was evaluated under the categories of color, pH, viscosity, and stability. Table 5 describes that the appearance of the formulation before and after heating–cooling cycles did not change. The viscosity of the base and MNP-loaded emulsion gel increased, probably caused by the evaporation of water in the formulations during the stability testing, but did not differ significantly. The % DPPH scavenging of the 1 g of MNPs containing the emulsion gel at day 0 was 85.35 ± 2.34%, while after heating–cooling, the MNP emulsion gel showed 93.81 ± 3.74%. The evaporation of water in the formulation after the heating–cooling test caused it to thicken, enhancing the mangiferin concentration. Moreover, Eudragit^®^ L100 is a delayed-release polymer; therefore, the % DPPH scavenging of the emulsion gel after heating–cooling stability presented as higher than at day 0. The result indicates that the emulsion gel containing MRE-loaded nanoparticles is stable.

### 3.8. Skin-Retention Study

A test was conducted to determine whether the emulsion gel containing MNPs could penetrate the Strat-M^®^ membrane, an artificial membrane that mimics human skin. The test results showed that no mangiferin content was found in the receptor fluid of the MRE solution and that of the emulsion gel containing MNPs at 30 min, and 1, 3, 6, 9, and 12 h, which indicated that the MRE and MNPs did not cross the Strat-M^®^ membrane and eventually appeared in the bloodstream. It suggests that the MNP emulsion gel is unlikely to have any systemic side-effects. It also has been shown that mangiferin accumulated more in the MNP-emulsion-gel membranes than in the MRE solution membranes, as presented in Figure 9. It indicates that the MNP emulsion gel helps mangiferin to penetrate the skin.

## 4. Conclusions

The MRE from the mango leaves of the “Talapnak” variety, which are otherwise considered as an agricultural waste material after harvesting, proved to have beneficial antioxidant and anticollagenase actions. From the proteomic analysis, it was demonstrated that the MRE is mostly involved in actin-filament organization. The MRE also possesses catalytic activity, which is consistent with its anticollagenase and antielastase activities. The emulsion gel containing electrosprayed nanoparticles loaded with the MRE was found to retain in the skin and be stable after six heating–cooling cycles. Electrosprayed nanoparticles containing the MRE from mango leaves may thus be a promising source of cost-effective cosmeceutical active ingredients to slow down skin-aging processes.

## Figures and Tables

**Figure 1 nanomaterials-13-02931-f001:**
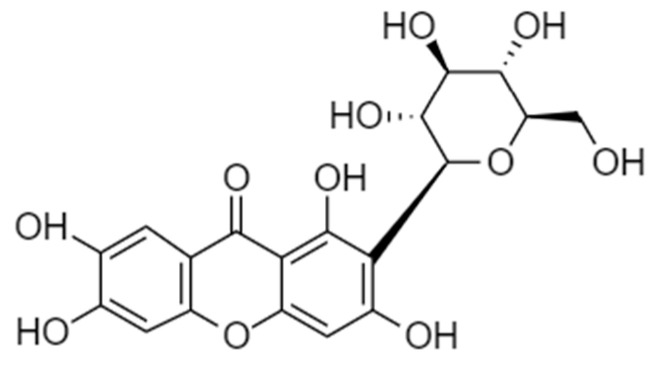
Chemical structure of mangiferin.

**Figure 2 nanomaterials-13-02931-f002:**
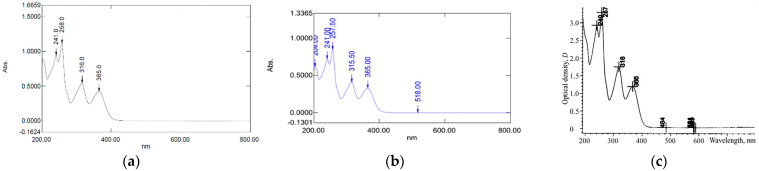
UV spectrum of the mangiferin–rich extract (MRE), (**a**) purified MRE (pMRE), (**b**) and the mangiferin spectrum presented in the Vietnamese pharmacopeia (**c**).

**Figure 3 nanomaterials-13-02931-f003:**
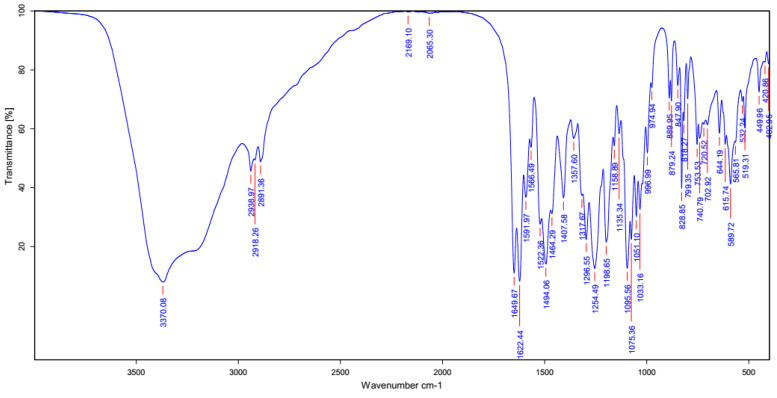
FTIR spectrum of the pMRE.

**Figure 4 nanomaterials-13-02931-f004:**
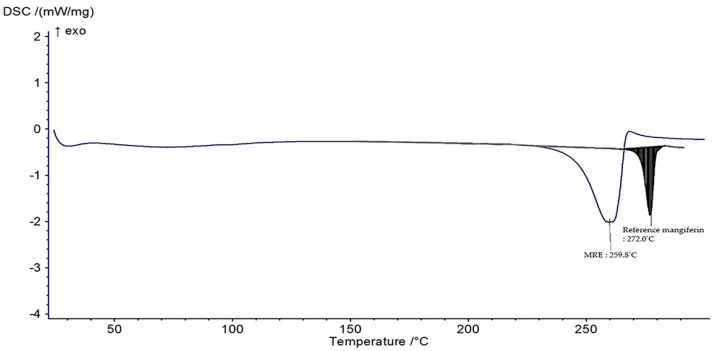
DSC thermogram of the MRE and reference mangiferin.

**Figure 5 nanomaterials-13-02931-f005:**
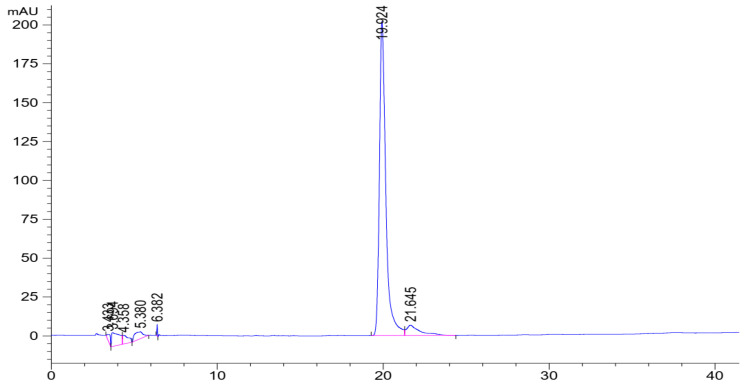
High-performance liquid chromatography chromatogram of the MRE.

**Figure 6 nanomaterials-13-02931-f006:**
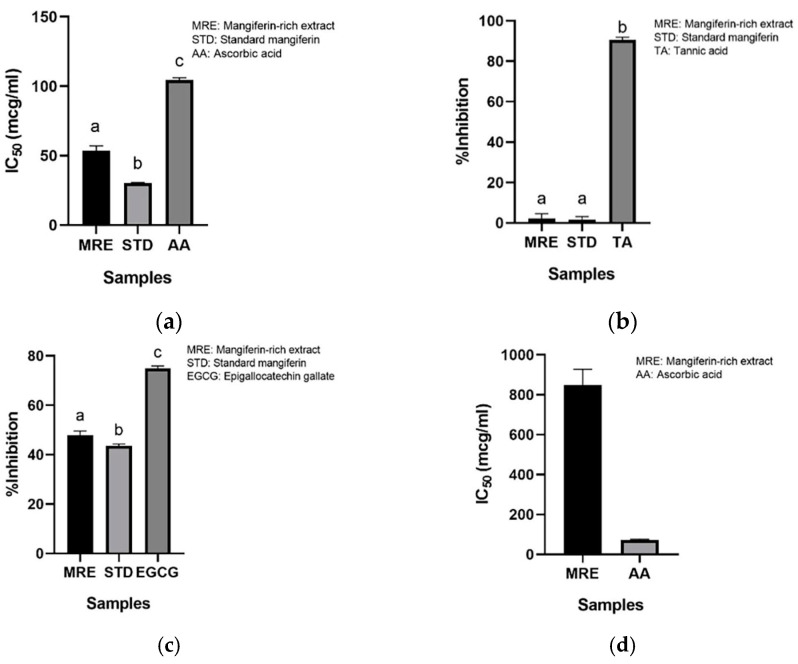
Anticollagenase activity of the MRE; (**a**) %Hyaluronidase inhibition of the MRE; (**b**) %Elastase inhibition of the MRE; (**c**) Antityrosinase activity of the MRE (**d**). All data are represented as mean ± SD (*n* = 3) and different alphabets indicate a significant difference.

**Figure 7 nanomaterials-13-02931-f007:**
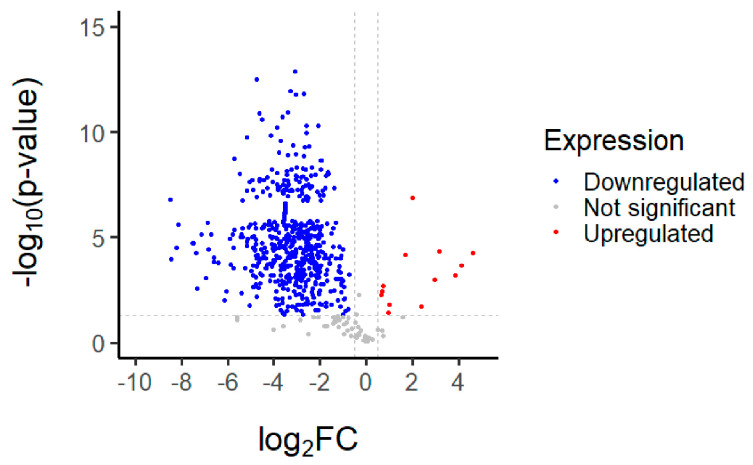
Volcano plot of the MRE analyzed using R-studio.

**Figure 8 nanomaterials-13-02931-f008:**
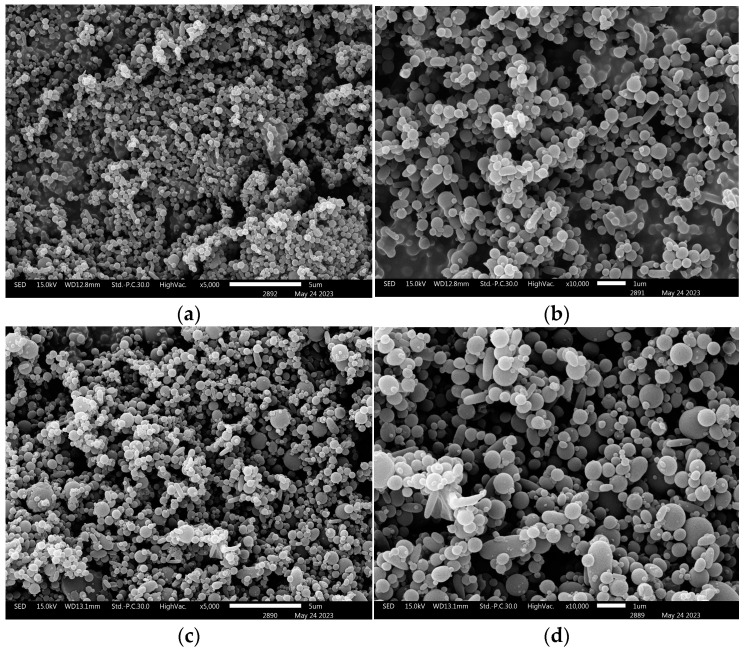
SEM images of the base polymer nanoparticles at 5000× (**a**) and 10,000× (**b**), and the MRE-loaded nanoparticles at 5000× (**c**) and 10,000× (**d**).

**Figure 9 nanomaterials-13-02931-f009:**
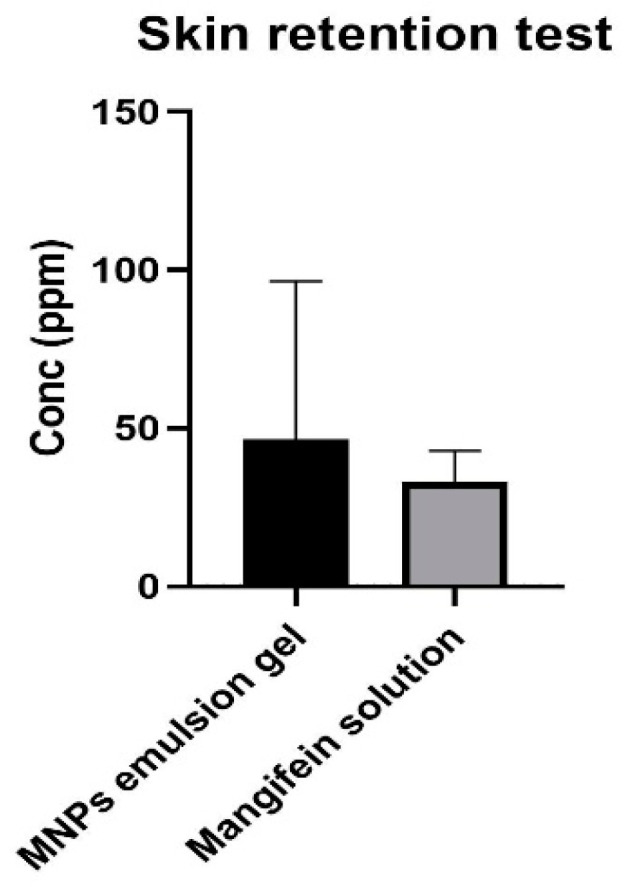
Mangiferin accumulation in the membranes of the skin-retention study. Data are represented as mean ± SD (*n* = 3).

**Table 1 nanomaterials-13-02931-t001:** MNPs containing emulsion-gel formulation.

Phase	Ingredient	Function	% *w*/*w*
A	DI water	Solvent	79.13
	Glycerin	Humectant	1.5
	Carbopol 940	Gelling agent	0.25
	MRE-loaded nanoparticles	Active agent	0.05
B	Sodium Polyacrylate (and) Polyisobutene (and) Water (Light-cream maker)	Polymeric emulsifier	1
	Jojoba oil	Emollient	0.5
C	Cyclopentasiloxane	Emollient	3
	Dimethicone	Emollient	1
	Cyclopentasiloxane and dimethicone cross polymer (DC 9045)	Slip modifier	3
D	Disodium EDTA in DI water	Chelator	0.02 10
E	DMDM hydantoin	Preservative	0.5
F	Triethanolamine	pH adjuster	0.1

**Table 2 nanomaterials-13-02931-t002:** Mangiferin content in the random samples.

Sample	Mangiferin Content (mg/g)
1	528.62 ± 41.50 ^a^
2	581.08 ± 27.89 ^a^
3	509.23 ± 28.74 ^a^

Data were expressed as mean ± SD (*n* = 3). ^a^ indicate a significant difference.

**Table 3 nanomaterials-13-02931-t003:** Top 11 potential functions of the upregulated proteins identified in the MRE compared with the control.

GO Term	Functions	False Discovery Rate
GO:0006936	Actin-filament organization	8.33 × 10 ^−5^
GO:0005200	Positive regulation of cytoskeleton organization	0.0103
GO:0051015	Rho protein signal transduction	0.0448
GO:0005198	Regulation of actin-filament polymerization	0.0448
GO:0044877	Cytoplasmic translation	0.0478
GO:0006936	Muscle contraction	0.0478
GO:1902905	Positive regulation of supramolecule fiber organization	0.0478
GO:0006412	Translation	0.0478
GO:0032956	Regulation of actin-cytoskeleton organization	0.0478
GO:0051493	Regulation of cytoskeleton organization	0.0478
GO:0050790	Regulation of catalytic activity	0.0478

**Table 4 nanomaterials-13-02931-t004:** Hen’s egg chorioallantoic membrane before and at 5 min.

Tested Substance	Before	At 5 min
Positive control (1% *w*/*v* SLS)	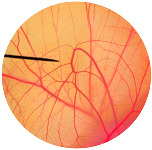	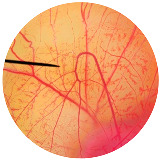
	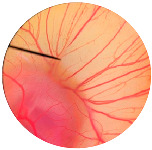	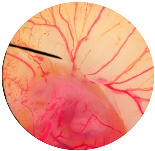
Negative control/Vehicle control (0.9% *w*/*v* NaCl)	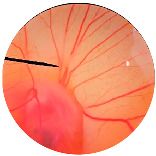	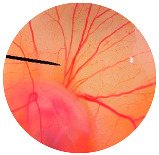
	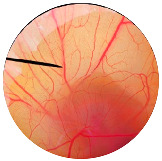	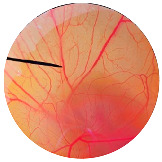
MNPs (0.5 mg/mL)	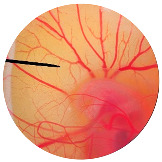	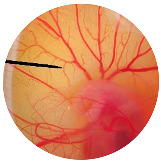
	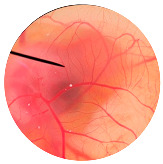	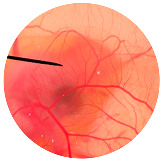
MRE (0.5 mg/mL)	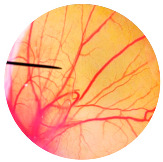	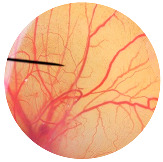
	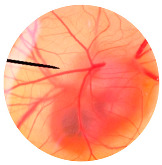	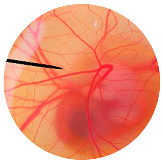

**Table 5 nanomaterials-13-02931-t005:** Stability of emulsion gel in heating–cooling cycles.

Formulation	Condition	Appearance	pH	Viscosity (Pa. s)	% DPPH Inhibition
Emulsion gel base	Day 0	White, opaque, no precipitation, homogenous	6.04	1.639 ± 0.05	ND
	H-C 6 cycles	White, opaque, no precipitation, homogenous	6.06	1.826 ± 0.09	
Emulsion gel containing MNPs	Day 0	Pale greenish, opaque, no precipitation, homogenous	6.07	1.968 ± 0.09	85.35 ± 2.34%
	H-C 6 cycles	Pale greenish, opaque, no precipitation, homogenous	6.07	2.020 ± 0.07	93.81 ± 3.74%

ND = not done.

## Data Availability

Data are contained within the article.

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
