# Peer review of "Electrosprayed Nanoparticles Containing Mangiferin-Rich Extract from Mango Leaves for Cosmeceutical Application"

_nanomaterials, 2023, doi:10.3390/nano13222931_

Round 1
Reviewer 1 Report
Comments and Suggestions for Authors
In this study, authors formulated the Mangiferin extract nanoparticles. Authors evaluated the powders using FTIR, UV spec, DSC, DPPH, anti-aging activity and proteomics analysis. Following are my suggestions and queries:
Line 68-78: Authors can cite some of the previous studies conducted with electrospraying for Mangiferin extract.
Authors can give a brief and purpose of the anticollagenase, antihyaluronidase, antielastase, and antityrosinase essays.
What is the purpose of purified MRE (pMRE)? Because many analysis was conducted with only MRE including the electrospraying.
Authors can explain the observance of FTIR and DSC analysis. Authors can discuss with the published reports.
Authors can expand the full form for AA (Ascorbic acid) and TA (Tannic acid) in the figure captions.
Section 3.3.2 to 3.3.5: Authors can add discussion with previous articles.
Figure 11: Authors can mention the scale bar in the figure caption.
Why bioactivity assays was conducted only for MRE? From the manuscript title, one can understand that this work is focused on the electrosprayed nanoparticles of Mangiferin-rich extract. But in this paper, only emulsion gels and skin retention analysis was conducted on the nanoparticle formulation.
Author Response
Dear Reviewers,
Thank you very much for taking the time to review this manuscript. Please see the attachment of detailed response.
Sincerely yours,
Author

Reviewer 2 Report
Comments and Suggestions for Authors
In your purified MRE, purity is around 50%, What else for other 50%?
Other 50% are similar to MRE or very different class?
Why melting point of your sample had around 10 celcius degree difference with standard compound?
Comments on the Quality of English Language
good and ready for publication
Author Response
Dear Reviewers,
Thank you for your kind review and constructive comments. Please see the attachment of detailed response.
sincerely yours,
Author

Reviewer 3 Report
Comments and Suggestions for Authors
Authors proposed a paper entitled “Electrosprayed Nanoparticles Containing Mangiferin-rich Extract from Mango Leaves for CosmeceuticalApplication” for the publication in Nanomaterials.
The paper has a good scientific soundness, and deserves to be published after major revisions.
I suggest adding an abbreviation list, according to the guidelines of this journal, such as MRE, etc.
Here is the list of my issues:
Line 19. “an important economic” please quantify
Line 30. “
The solution was sprayed with applied voltage 12 kV, flow rate 0.7mL/h and” I suggest limiting at the description of results. This sentence needs to be moved or only left to methods.
Line 32. “The %entrapment efficiency presented 84.9%” please say “the entrapment efficiency was quantified/measured…”
Line 40 “ Anacardiaceae family”. Italics?
Line 40 “ Ayurvedic” why capital letter?
Line 51. Why citing specific vitamins and then generic antiviral, anticancer antidiabetic etc?
Line 77. “(%EE)” in my opinion, “%” is useless here.
Line 122. “UV-VIS -“ should be “UV-Vis”
Line 155. Why using bold in this formula? Please uniform formula according to the guidelines of this journal. Do they need to be numbered and cited in the manuscript?
Line 160. “ported as mean ± SD.2.5.1.2” I believe that this is a mistake. Please correct it.
Line 182. “0.03% hyaluronic acid was added before” was this on mass basis?
Line 201. The formula here is exactly the same as above. Please number the previous and eliminate this. Recall this formula number if necessary. Declaration of the same variables is not needed again.
Line 254. Please add references to support the method expressed in this paragraph.
Table 1, line 2. “0.05%”, please remove “%”.
Line 306. “70% isopropanol gave the greatest yield compared” I would say that the use of 70% isopropanol…”
Figure 2. Please increase the dimensions of these diagrams.
Table 2. Useless to repeat the word “sample” in each line of this table.
Figure 6. Title is not necessary here: better using the caption.
Figure 7. As above.
Figure 6 to 9. Why do not imagine to create an unique composition of figures from 6a to 6d?
Figure 10 maybe should be characterized by square dimensions.
Figure 11. Why overlapping letters on the figures? Please remove them and put them on the sides of the figures.
Table 5. “S” should not be capital letter.
Table 5. I would not say “% DPPH”, I would say “inhibition of the antioxidant power”. Please find an abbreviation for this: DPPH is just the molecule.
Comments on the Quality of English Language
Moderate revisions of English suggested.
Author Response

(The authors gave the same response as above.)

Round 2
Reviewer 1 Report
Comments and Suggestions for Authors
Authors responded all the queries and significantly modified the manuscript.
Author Response
Dear reviewer,
Thank you to all of your comments to this manuscript. The response to your feedback is attached.
Sincerely yours,
Author

Reviewer 3 Report
Comments and Suggestions for Authors
Authors significantly improved their manuscript after a proper revision.
Authors also followed my points, but only two issues still need to be addressed:
Issue Nr 1.
I cannot the the abbreviation list. Did the author add it?
Issue Nr. 2
“I have edited “Mango is one of the most important economic fruits.” On Page 1, Line 19. According to the information of trade policy and strategy office of Thailand, In 2021, Thailand exported mangos around113,806 tons valuing 2.9 billions baths. The expanding rate jumped to 50.25% higher than the previous month.”
Authors did not add this numeric information in the manuscript.
Author Response
Dear reviewer,
Thank you to all of your comments to this manuscript. The response to your feedback is attached
Sincerely yours,
Author
